# Sparse but Sharp: Saliency-guided Black-box Attacks Reveal Vulnerabilities in Skeleton-based Human Action Recognition

## Abstract

The remarkable performance of Graph Convolutional Networks and Transformers in skeleton-based human action recognition is a progress to celebrate. However, recent studies reveal their vulnerability to adversarial attacks. We focus on black-box attacks, which hold greater practical relevance, and propose the first saliency-guided spatio-temporal sparse black-box attack framework for skeleton-based recognition. By estimating the contribution of joints and frame segments to recognition accuracy, our solution is able to inject perturbations only into a localised set, thereby enhancing stealth. Compatible with both confidence-based and label-only black-box settings, our framework offers broad applicability in real-world scenarios. We conduct a comprehensive evaluation of the proposed attack methodology on public large-scale datasets and compare its performance with SOTA algorithms. It is demonstrated that our attack strategy achieves competitive or even superior effectiveness in most settings, while offering better imperceptibility and a favourable balance between query efficiency and attack performance. Importantly, our evaluation reveals significant disparities in robustness across existing action recognition models. Our solution presents a practical paradigm for efficient sparse attack strategies, providing novel insights into the structural robustness of skeleton-based recognition methods.

## 1 Introduction

Skeleton-based human action recognition has attracted considerable attention Ren et al. (2024); Feng et al. (2022) in recent years due to the strong robustness of skeletons against background noise, illumination changes, and the powerful ability to model human action. It has been widely applied in various domains, including anomaly detection, VR / AR, and human-computer interactionNikam & Ambekar (2016); Nwakanma et al. (2021); Yang et al. (2019). With the rapid advancement of deep learning, an increasing number of solutions for skeleton-based action recognition have been developed based on Graph Convolutional Networks (GCNs) and Transformers Yan et al. (2018); Cheng et al. (2020); Plizzari et al. (2021). However, recent studies have shown that despite their excellent performance, these methods remain highly vulnerable to adversarial samples Liu et al. (2020a); Wang et al. (2021); Zheng et al. (2020). These adversarial samples are generated by adding subtle and humanly-imperceptible perturbations to the input samples, resulting in significant misclassification during the inference stage Szegedy et al. (2013); Goodfellow et al. (2014). Such vulnerabilities pose a significant threat to the deployment of action recognition systems in safety-critical applications such as surveillance, autonomous systems, and healthcare Akhtar & Mian (2018); Carlini & Wagner (2017).

Compared with images or conventional time-series data, skeleton data exhibits unique structural characteristics. Although the data representation may involve tens of thousands of spatio-temporal dimensions, the physical structure of the human body and physiological constraints introduce substantial redundancy into the sequences Lin et al. (2020). This redundancy implies that the models could be built by relying just on a few key frames and joints during recognition, thus providing a theoretical foundation for sparsity-based adversarial attacksDai et al. (2023). Conversely, suppose an adversarial attacker can successfully mislead the model by perturbing only a few joints in a lim-

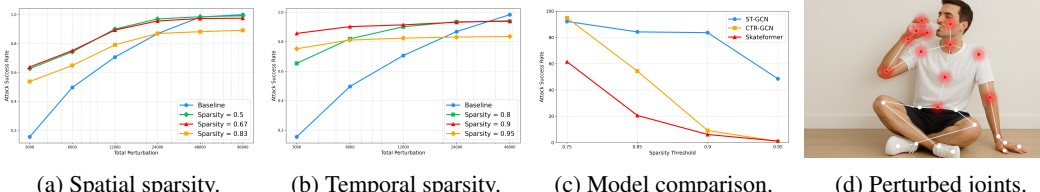

(a) Spatial sparsity.     (b) Temporal sparsity.     (c) Model comparison.     (d) Perturbed joints.

Figure 1: Intuitive insight into the performance of our spatio-temporal sparse attacking solution. (a) presents the impact of spatial sparsity, where perturbations are confined to key joint regions. (b) shows the results produced by temporal sparsity where perturbations are applied selectively across frame segments. Both (a) and (b) report targeted attack success rates on ST-GCN evaluated on the NTU60 dataset. (c) compares the targeted attack success rates across different victim architectures (ST-GCN, CTR-GCN, and SkateFormer) under varying sparsity levels on NTU60. (d) illustrates the spatial positions of the perturbed joints on an action example.

ited number of frames. In that case, it suggests that the recognition of models is highly dependent on the localised features.

Current research on the adversarial robustness of skeleton-based action recognition models remains relatively limited, with black-box settings receiving particularly little attention Wang et al. (2021). To the best of our knowledge, BASAR is the only implemented query-based black-box attack method specifically designed for skeleton-based action recognition Diao et al. (2021). While BASAR achieves effective performance, it applies global perturbations across the entire sequence, resulting in a large number of modified variables and significant computational overhead. In contrast, transfer-based black-box methods such as TASAR Diao et al. (2024) avoid sending queries directly by attacking surrogate models, but they often suffer from limited transferability and suboptimal success rates when applied to target models.

Therefore, a core challenge lies in minimising both the number of perturbed variables and the number of model queries, while maintaining a high attack success rate. We argue that introducing the principle of a sparsity-aware adversarial attack is critical. Rather than applying indiscriminate perturbations across the sequence, the attacker should be guided by an analysis of the model's sensitivity to different frames and joints. By localising and perturbing only the most critical spatio-temporal parts, the attack can achieve higher stealth and attack efficacy.

This paper proposes a heuristic spatio-temporally sparse black-box adversarial attack method tailored for skeleton-based action recognition. To evaluate the practicality of the proposed method under realistic adversarial constraints, we conducted comprehensive experiments on several mainstream skeleton-based action recognition solutions under varying perturbation budgets. The results demonstrate that although the baseline approach, which applies dense perturbations over the entire sequence, achieves a nearly 100% attack success rate, it heavily relies on high-intensity interference and lacks stealth, rendering it unsuitable for deployment in security-sensitive scenarios. In contrast, our sparsity-aware black-box attack achieves comparable, or even higher success rates, while significantly reducing perturbation intensity across most configurations. This finding underscores the unique advantages of sparse strategies in low-intervention and high-impact attacks. By precisely identifying and perturbing the most decision-critical frames and joints, our method successfully deceives the victim models under black-box, gradient-free, and highly constrained conditions, greatly enhancing the practicality and deployability of adversarial attacks.

In summary, the main contributions of this paper are as follows:

- We propose a novel sparse black-box adversarial attack framework tailored for skeleton-based action recognition, which jointly exploits temporal and spatial sparsity during the attack process.

- We introduce a saliency-guided joint selection strategy that effectively locates the most influential spatio-temporal regions without requiring access to model gradients.

- We conduct an in-depth analysis of the adversarial vulnerabilities of both GCN-based and Transformer-based models, revealing their differing sensitivities against spatio-temporal perturbations and providing new insights for future defence strategies.

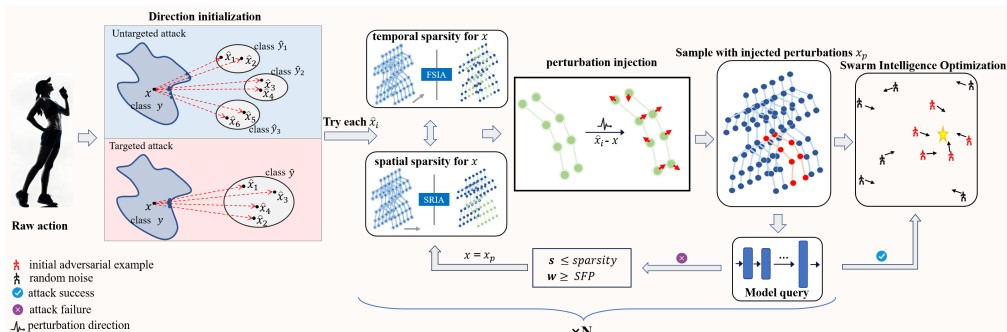

Figure 2: The overall framework of the proposed sparse black-box attack strategy. Starting from a raw action sequence, the attacker first performs direction initialisation using samples from the target class, generating multiple candidate perturbation directions $\hat{x}_i - x$. The perturbations are then iteratively injected along each direction, guided by both temporal and spatial sparsity constraints to restrict modifications to salient frames and joint regions, yielding an initial adversarial example $x_p$. After each injection, model queries are submitted for evaluation, and failed attempts trigger re-selection of both the direction and the region. Finally, swarm intelligence optimisation is applied to refine the adversarial sample, while preserving a natural skeletal structure, thereby improving attack success rate and imperceptibility. Here, $s$ denotes the minimum required sparsity, and $w$ represents the single-frame perturbation threshold.

Our study not only presents a practical adversarial attack strategy for skeleton-based action recognition but also contributes theoretical insights, as well as a better understanding of deep model decision-making mechanisms, thus enhancing their robustness in safety-critical applications.

## 2 RELATED WORK

**Skeleton-Based Action Recognition** Skeleton-based action recognition has progressed from handcrafted features Devanne et al. (2013); Fernando et al. (2015); Vemulapalli et al. (2014) to deep neural models. Early deep methods adopted Recurrent Neural Networks to capture temporal dynamics Du et al. (2015); Zhang et al. (2017); Song et al. (2017), but cannot perform spatial structural modelling well. Graph Convolutional Networks (GCNs) were introduced by ST-GCN Yan et al. (2018) to overcome this deficiency. They represent skeletons as spatio-temporal graphs, thus initiating a new paradigm. Subsequent GCN-based approaches expanded this framework: AS-GCN Li et al. (2019) adds action-conditioned topologies, 2s-AGCN Shi et al. (2019) uses dual streams for the joint and bone features, CTR-GCN Chen et al. (2021) models channel-wise topologies, InfoGCN Chi et al. (2022) incorporates contrastive learning, and BlockGCN Zhou et al. (2024) restructures global graph design. In parallel, Transformer-based models emerged with ST-TR Plizzari et al. (2021), Motionformer Patrick et al. (2021), and PoseFormer Zheng et al. (2021), which leverage self-attention to model long-range dependencies. More recently, hybrid architectures combining GCNs and Transformers, such as DSTA-Net Shi et al. (2020), STGT Liu et al. (2020b), and Skate-Former Do & Kim (2024), have demonstrated the merits of integrating structural priors with global attention, reflecting the research shift toward unified spatio-temporal modelling.

**Adversarial Attacks on Skeleton-Based Action Recognition** The susceptibility of image and video-based action recognition models to adversarial attacks has been widely studied. White-box attacks Madry et al. (2017); Moosavi-Dezfooli et al. (2016); Papernot et al. (2016b); Wei et al. (2019); Mu et al. (2021), black-box query-based attacks Jiang et al. (2019); Li et al. (2021); Papernot et al. (2016a); Brendel et al. (2017); Wei et al. (2020), and transfer-based attacks Liu et al. (2016); Gu et al. (2023); Wei et al. (2022a;b); Dong et al. (2018) reveal model vulnerabilities under varied access settings. These findings have spurred interest in studying the robustness of skeleton-based action recognition models. Though skeleton-based methods achieve strong performance, they remain highly vulnerable to adversarial perturbations. In white-box settings, CIASA Liu et al. (2020a) and SMART Wang et al. (2021) leverage structural and perceptual priors to craft imperceptible, temporally consistent attacks. Structured variants like bone length perturbations Tanaka et al. (2022) and

sparse metaverse attacks Dai et al. (2023) further validate their vulnerability to localised manipulations. Yet the assumptions underlying white-box attacks rarely hold in practice. Black-box strategies have thus emerged. BASAR Diao et al. (2021) estimates gradients via sampling. Similarly, TASAR Diao et al. (2024) leverages surrogate models, and Hard No-Box Lu et al. (2023) eliminates queries using motion-informed priors. Despite their specific designs, most methods use dense, joint-agnostic perturbations, reducing stealth and efficiency. Given data redundancy and joint-level saliency of a skeleton sequence, we advocate a spatio-temporally sparse black-box attack paradigm balancing effectiveness, imperceptibility, and cost.

## 3 METHODOLOGY

In this section, we present a sparse adversarial attack framework tailored for skeleton-based action recognition. Let $\mathbf{x}_0 \in \mathbb{R}^{C \times T \times V \times M}$ denote the input skeleton sequence, where $C$, $T$, $V$, and $M$ represent the number of channels (*e.g.*, 3D coordinates), frames, joints, and actors, respectively. $\mathbf{x}_{\text{adv}}$ denotes the adversarial skeleton sequence obtained by adding perturbations to $\mathbf{x}_0$. We define the target model as a black-box classifier $F : \mathbb{R}^{C \times T \times V \times M} \to \mathcal{Y}$, which outputs the predicted class label $\hat{y} = F(\mathbf{x}_0)$ without exposing any internal structure or gradient information.

We consider two types of attack objectives Szegedy et al. (2013). The first is **untargeted attack**, the goal of which is to mislead the model's prediction from the true label $y$, *i.e.*, $F(\mathbf{x}_{\text{adv}}) \neq y$. The second is **targeted attack**, aiming to mislead the model into predicting a specific target class $y_{\text{adv}} \neq y$, such that $F(\mathbf{x}_{\text{adv}}) = y_{\text{adv}}$.

### 3.1 TEMPORAL SPARSITY

To improve the stealth and efficacy of adversarial attacks in skeleton-based action recognition, we propose a **temporal sparsity strategy** that selectively perturbs only a few critical frame segments to achieve highly deceptive effects under a constrained perturbation budget. In contrast to conventional methods that inject perturbations uniformly across the entire temporal sequence, we design a **Frame Segment Importance Assessment (FSIA)** mechanism to identify the most influential segments along the temporal dimension.

During initialisation, we select a reference sample $\hat{x}$ from the target class and construct a vector $\delta = \hat{x} - x_0$ as a perturbation guidance signal. The input sequence is partitioned into a set of candidate frame segments $S = \{s_1, s_2, \dots\}$, each with a fixed length $l$. For each candidate $s_i$, we

---

**Algorithm 1** Frame Segment Importance Assessment Mechanism under Targeted Attacks

---

1: **Input:** Targeted recogniser model $F$, Original skeleton sequence $x_0$, Target adversarial class label $y_{\text{adv}}$, Frame-segment length $l$, Sequence length $T$
2: **Output:** $M$ — a binary mask indicating key frame segment
3: $\hat{x} \leftarrow$ a skeleton sequence sample from target class $y_{\text{adv}}$
4: $p \leftarrow \hat{x} - x_0$               // Direction towards target class sample
5: $score\_dict \leftarrow \{\}$
6: **for** $i = 0$ **to** $T - l$ **do**
7:     $segment \leftarrow [i, i+1, \dots, i+l-1]$
8:     $M \leftarrow \text{ones\_like}(x_0)$
9:     $M[:, segment, :, :] \leftarrow 0$            // Mask this frame segment
10:     $x_{\text{temp}} \leftarrow x_0 + p \cdot M$
11:     $conf \leftarrow F.predict\_confidence(x_{\text{temp}}, y_{\text{adv}})$
12:     $score\_dict[i] \leftarrow conf$
13: **end for**
14: $best\_start \leftarrow \arg\min(score\_dict)$
15: $key\_segment \leftarrow [best\_start, \dots, best\_start + l - 1]$
16: $M \leftarrow \text{zeros\_like}(x_0)$
17: $M[:, key\_segment, :, :] \leftarrow 1$            // Set key segment to be preserved
18: **return** $M$

---

construct a masked version of the perturbed input as follows:

$$\mathbf{x}_i = \mathbf{x}_0 + \delta \odot \mathbf{M}_i, \quad \mathbf{M}_i(t) = \begin{cases} 0, & t \in s_i \\ 1, & \text{otherwise} \end{cases}. \tag{1}$$

Here, $\delta$ is a fixed perturbation vector shared across different segments $s_i$, and the mask $\mathbf{M}_i$ controls the range of perturbation applied, determining the regions where perturbations are applied.

We evaluate the importance of each frame segment by measuring the change in confidence for the target class after masking its associated perturbation. Segments that cause a significant confidence drop are considered critical for maintaining adversarial influence. The segment leading to the greatest reduction is selected as the key intervention region for subsequent perturbation optimisation.

To prevent deviation of the perturbation direction from the decision boundary, we further introduce a **perturbation direction refinement mechanism**, which iteratively adjusts $\delta$ over the identified key segments to enhance convergence and attack stability. The details of this mechanism are provided in Algorithm 2 in the appendix.

### 3.2 SPATIAL SPARSITY

In addition to temporal sparsity modelling, we further introduce a spatial sparsity strategy for skeletal data, aiming to enhance the local focus and perceptual imperceptibility of adversarial perturbations. The core idea is to partition human joints into structured regions and constrain the perturbations to be applied only to a highly discriminative subset of spatial regions based on their importance. We define two spatial partitioning strategies:

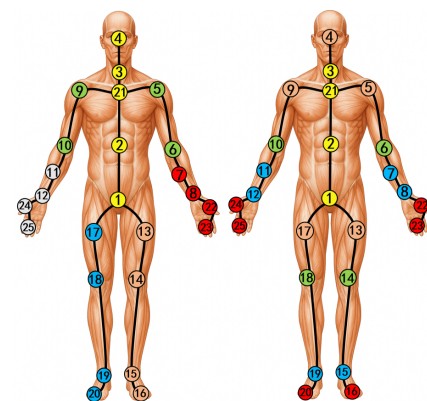

1. **Local-structure Partition**: Inspired by Shahroudy et al. (2016); Song et al. (2020), we divide the skeleton into six anatomically coherent local regions (*e.g.*, forearms, lower legs, torso) based on kinematic adjacency, enabling fine-grained capture of localised motion features.

2. **Global-saliency Partition**: Following the hierarchical decomposition principle in Lee et al. (2023), we centre the partition at the chest joint and group nodes by their radial distance. Distal joints—such as the head, fingertips, and toes—are clustered to emphasise semantically salient regions with high expressive value in action execution.

Figure 3: Two spatial partition strategies for selecting adversarial regions: the left shows the Local-structure Partition and the right shows the Global-saliency Partition.

The spatial perturbation selection and injection process follows the same design principle as the temporal sparsity strategy. Specifically, we systematically mask out combinations of spatial regions and evaluate their collective impact on the model's prediction. By observing the confidence changes resulting from each candidate group suppression, we identify the most influential regions for subsequent perturbation injection. Perturbations are then generated and optimised only within these selected regions.

### 3.3 OVERALL FRAMEWORK

To achieve joint sparsity of adversarial perturbations across temporal and spatial dimensions, we propose an end-to-end attack framework that aims to minimally perturb the input by precisely targeting the most critical fragments of the skeleton sequence. The framework leverages Frame Segment Importance Assessment (FSIA) and Spatial Region Importance Assessment (SRIA) to identify critical temporal segments and high-impact spatial regions, which are then integrated into a unified spatio-temporal mask.

Under this mask constraint, the system estimates the perturbation direction and incrementally applies it using a confidence-guided adaptive step-size strategy to generate the initial adversarial sample.

To avoid overly concentrated perturbations on individual frames, we introduce the Single-Frame Perturbation (SFP) Kennedy & Eberhart (1995); Holland (1992) metric to quantify the perturbation magnitude within each frame. If the SFP of any intermediate sample $x_{adv}$ exceeds a predefined threshold $w$, the attack is terminated to prevent excessive perturbation on specific frames, which could lead to local spikes or structural discontinuities. This mechanism helps ensure the imperceptibility and practical availability of the adversarial examples.

To further enhance the imperceptibility of the initial adversarial samples, we introduce a Swarm Intelligence Optimisation (SIO) strategy, which searches from multiple initial candidates to identify an optimal sample that achieves attack success with minimal perturbation. During optimisation, a bone-length constraint is imposed to preserve structural consistency and prevent unrealistic limb deformations. While maintaining the effectiveness of the attack, SIO significantly reduces the perturbation density across both spatial and temporal dimensions. The complete procedure is detailed in Algorithm 2 of the Appendix.

### 3.4 Attack under Highly Constrained Settings

To evaluate the practicality of sparse attacks in severely information-constrained environments, we extend our framework to a strict **label-only** black-box setting, where the attacker can access only predicted class labels without any confidence scores or softmax outputs, making the identification of critical regions significantly more challenging.

To enable effective identification of critical temporal segments and spatial regions under such constraints, we design a *hierarchical mask-based intervention mechanism*. Specifically, for each candidate frame segment $s$ or spatial region $\mathbf{r}$, we construct a set of spatio-temporal masks $\mathbf{M}^{(k)}(t)$, and apply perturbations of decreasing magnitudes to the selected region across multiple levels. The perturbation strengths are scaled using a predefined set of attenuation factors $\alpha_k \in \{0.8, 0.6, 0.4, 0.2, 0.0\}$.

In each mask, perturbations within the selected region are multiplied by $\alpha_k$, while all other locations remain unchanged. Each candidate adversarial example is thus constructed as:

$$\mathbf{x}_{\text{adv}}^{(k)} = \mathbf{x}_0 + \delta \odot \mathbf{M}^{(k)}(t),$$

$$\text{where } \mathbf{M}^{(k)}(t) = \begin{cases} \alpha_k, & t \in s \text{ or } v \in \mathbf{r}, \\ 1, & \text{otherwise.} \end{cases} \quad (2)$$

We record the attenuation level $\alpha_k$ at which the model's prediction first deviates from the target class. A larger $\alpha_k$ indicates that the region causes a label change under a weaker perturbation, implying higher sensitivity.

Finally, we rank all candidate regions based on their sensitivity and inject perturbations into the most critical ones. Due to the absence of confidence feedback, the perturbation direction cannot be iteratively refined, which may lead to slightly reduced convergence and controllability. Nevertheless, experiments show that the proposed mechanism remains robust under hard-label settings, demonstrating strong adaptability to severely constrained environments.

## 4 Evaluation

### 4.1 Experimental Setting

**Datasets.** We evaluate our method on two widely used skeleton-based action recognition datasets: NTU RGB+D 60 Shahroudy et al. (2016) and NTU RGB+D 120 Liu et al. (2019). NTU60 contains 60 action classes, while NTU120 expands to 120 classes with increased subject and viewpoint diversity. Both datasets offer temporal 3D joint sequences, making them well-suited for analysing spatio-temporal adversarial perturbations.

**Target Models.** We consider three representative models: ST-GCN Yan et al. (2018), CTR-GCN Chen et al. (2021), and SkateFormer Do & Kim (2024), covering both GCN- and Transformer-

Table 1: Attack performance under different sparsity levels in the temporal domain only.

| (a) Untargeted Attack | | | | |
|---|---|---|---|---|
| **Min.$s$** | **Setting** | **QC** | **$\ell_2$** | **ASR (%)** |
| 0.0 | UA | 2097.57 | 0.10 | 100.00 |
| 0.9 | UA | 11386.04 | 0.04 | 99.84 |
| 0.95 | UA | 14369.64 | 0.03 | 98.96 |
| 0.99 | UA | 16550.17 | 0.02 | 93.76 |
| 0.997 | UA | 21296.28 | 0.01 | 90.96 |

| (b) Targeted Attack | | | | |
|---|---|---|---|---|
| **Min.$s$** | **Setting** | **QC** | **$\ell_2$** | **ASR (%)** |
| 0.0 | TA | 3895.79 | 0.08 | 95.32 |
| 0.9 | TA | 3671.37 | 0.06 | 83.60 |
| 0.95 | TA | 3246.54 | 0.04 | 64.59 |
| 0.99 | TA | 4756.21 | 0.03 | 16.42 |
| 0.997 | TA | 10746.06 | 0.02 | 5.60 |

Table 2: Attack performance under different sparsity levels in the spatial domain only.

| (a) Local-structure Partition | | | | |
|---|---|---|---|---|
| **Min. $s$** | **Setting** | **QC** | **$\ell_2$** | **ASR (%)** |
| 0.5 | UA | 4354.55 | 0.04 | 100.00 |
| | TA | 4801.34 | 0.12 | 93.91 |
| 0.67 | UA | 4758.85 | 0.03 | 100.00 |
| | TA | 4542.96 | 0.10 | 85.30 |
| 0.83 | UA | 5388.93 | 0.02 | 99.93 |
| | TA | 2191.31 | 0.02 | 1.93 |

| (b) Global-saliency Partition | | | | |
|---|---|---|---|---|
| **Min. $s$** | **Setting** | **QC** | **$\ell_2$** | **ASR (%)** |
| 0.4 | UA | 4256.63 | 0.03 | 100.00 |
| | TA | 3111.89 | 0.10 | 85.45 |
| 0.6 | UA | 4540.40 | 0.02 | 100.00 |
| | TA | 3175.57 | 0.08 | 59.69 |
| 0.8 | UA | 5066.60 | 0.01 | 99.93 |
| | TA | 2199.21 | 0.02 | 2.15 |

based architectures. These models are widely adopted and serve as strong benchmarks for evaluating the effectiveness and transferability of our sparse attack framework.

**Metrics.** We report three evaluation metrics to comprehensively assess attack performance: (1) Attack Success Rate (ASR) — the percentage of adversarial examples that successfully change the model's prediction; (2) Query Count (QC) — the average number of model queries required to generate a successful adversarial sample; and (3) $l_2$ — the average perturbation magnitude applied to each frame, measuring the stealthiness of the attack. The $l_2$ calculation is defined as follows:

$$l_2 = \frac{1}{nN} \sum_{j=1}^{N} \|x^{(j)} - x'^{(j)}\|_2, \tag{3}$$

where $N$ is the number of adversarial samples, $n$ is the number of frames per sample, and $x^{(j)}$ and $x'^{(j)}$ represent the original sample and its corresponding adversarial sample, respectively.

## 4.2 ABLATION STUDY: TEMPORAL AND SPATIAL SPARSITY

We systematically evaluate the impact of temporal and spatial sparsity strategies on adversarial performance using the NTU60 Shahroudy et al. (2016) dataset and ST-GCN Yan et al. (2018) as the target model. All experiments are conducted under a fixed single-frame perturbation threshold of $w = 100$, covering both targeted and untargeted attack settings. The results are presented in Table 1 and Table 2.

We introduce a sparsity threshold $s$ to restrict perturbations to regions with sparsity $\geq s$, where the **baseline case** ($s = 0$) corresponds to an unconstrained attack.

The results show that the baseline achieves higher attack success rates (ASR) due to its larger optimisation space, but its perturbations are more widely distributed, reducing stealth. As sparsity increases, ASR drops more significantly in targeted attacks, as the constrained search space makes it harder to guide predictions toward the target class. In contrast, untargeted attacks exhibit stronger robustness to sparsity constraints.

Moreover, higher sparsity leads to increased query counts (QC), but also significantly reduces the per-frame perturbation magnitude (measured by $\ell_2$ norm), thereby improving the imperceptibility of adversarial examples.

Table 3: Untargeted and targeted attacks against skeleton-based models on NTU datasets.

| Dataset | Target Model | Attack Model | Untargeted Attacks | | | | Targeted Attacks | | | |
|---|---|---|---|---|---|---|---|---|---|---|
| | | | QC | $\ell_2$ | S | ASR(%) | QC | $\ell_2$ | S | ASR(%) |
| NTU60 | ST-GCN | Baseline | 1044.99 | 0.01 | 0.00 | 33.74 | 1037.55 | 0.02 | 0.00 | 15.44 |
| | | (T1. + S1.) | 14606.60 | 0.02 | 0.95 | 99.85 | 5640.74 | 0.03 | 0.95 | 61.25 |
| | | (T1. + S2.) | 14399.29 | 0.02 | 0.94 | 100.0 | 12957.38 | 0.05 | 0.94 | 85.15 |
| | CTR-GCN | Baseline | 1103.53 | 0.09 | 0.00 | 82.64 | 1126.36 | 0.10 | 0.00 | 78.77 |
| | | (T3. + S1.) | 9643.62 | 0.08 | 0.90 | 99.20 | 11821.72 | 0.23 | 0.90 | 85.02 |
| | | (T2. + S3.) | 8543.20 | 0.07 | 0.90 | 99.63 | 10921.09 | 0.16 | 0.90 | 93.56 |
| | SkateFormer | Baseline | 1181.67 | 0.11 | 0.00 | 97.72 | 1183.34 | 0.12 | 0.00 | 95.03 |
| | | (T1. + S1.) | 10956.31 | 0.10 | 0.84 | 98.69 | 6001.20 | 0.10 | 0.84 | 61.46 |
| | | (T1. + S2.) | 9526.06 | 0.11 | 0.81 | 99.46 | 5924.45 | 0.12 | 0.81 | 96.48 |
| NTU120 | ST-GCN | Baseline | 3215.83 | 0.02 | 0.00 | 41.53 | 5326.33 | 0.04 | 0.00 | 5.41 |
| | | (T1. + S1.) | 20427.29 | 0.15 | 0.95 | 99.35 | 20181.79 | 0.64 | 0.95 | 87.54 |
| | | (T1. + S2.) | 25795.98 | 0.17 | 0.94 | 94.46 | 19958.69 | 0.50 | 0.94 | 91.97 |
| | CTR-GCN | Baseline | 1059.93 | 0.07 | 0.07 | 67.09 | 1148.48 | 0.07 | 0.00 | 69.11 |
| | | (T3. + S1.) | 8484.47 | 0.08 | 0.90 | 98.66 | 6778.41 | 0.20 | 0.90 | 45.07 |
| | | (T2. + S3.) | 7573.59 | 0.12 | 0.90 | 99.53 | 6540.31 | 0.25 | 0.90 | 52.71 |
| | SkateFormer | Baseline | 1239.31 | 0.11 | 0.00 | 92.32 | 1276.32 | 0.13 | 0.00 | 93.35 |
| | | (T1. + S1.) | 8438.78 | 0.10 | 0.84 | 98.42 | 5953.58 | 0.16 | 0.84 | 20.83 |
| | | (T1. + S2.) | 8038.49 | 0.12 | 0.81 | 99.11 | 5936.68 | 0.21 | 0.81 | 68.53 |

These trends are further illustrated in Fig. 1a and Fig. 1b. Under equal perturbation budgets, sparse attacks achieve better trade-offs between effectiveness and stealth compared to the baseline. Notably, temporal sparsity consistently outperforms spatial sparsity under low-budget scenarios, suggesting that ST-GCN is more vulnerable to temporal perturbations—potentially due to limitations in its temporal modelling capabilities.

## 4.3 SPATIO-TEMPORAL ATTACK V.S. BASELINE

This section compares the performance of the baseline method and our proposed spatio-temporal sparse attack strategies on the NTU60 Shahroudy et al. (2016) and NTU120 Liu et al. (2019) datasets. All methods are executed under the same total perturbation budget when attacking the same model on the same dataset to ensure fairness, as summarised in Table 3.

It is worth noting that the baseline method is derived from our attack framework by removing the sparsity constraints, allowing perturbations to be freely distributed across the entire input sequence. As such, it serves as a fully unconstrained attack and provides a reference lower bound for evaluating

For ST-GCN, which processes the full 300-frame sequences, we set the total perturbation budgets to 3000 and 15000 for NTU60 and NTU120, respectively. The higher budget on NTU120 accounts for its larger scale and more diverse class set, which significantly increases the difficulty of targeted attacks. For CTR-GCN and SkateFormer, the budget is fixed at 1350 across both datasets to facilitate direct performance comparison across models.

In detail, T1, T2, and T3 denote the selection of 30, 15, and 12 frames in the temporal dimension. S1 corresponds to selecting 3 regions under the Local-structure Partition, while S2 and S3 select 3 and 2 regions, respectively, under the Global-saliency Partition.

Experimental results show that across all three models—ST-GCN Yan et al. (2018), CTR-GCN Chen et al. (2021), and SkateFormer Do & Kim (2024)—our method consistently outperforms the baseline in most configurations. In particular, on ST-GCN, sparse attacks significantly improve the attack success rate (ASR) for both targeted and untargeted settings on NTU60 and NTU120 (*e.g.*, T1+S2 achieves 91.97% on NTU120, compared to only 5.41% for the baseline). For CTR-GCN, our approach also demonstrates advantages on NTU60, while showing slightly lower ASR on NTU120 in some settings, possibly due to the limited sparsity budget. SkateFormer exhibits inherently stronger structural robustness, where our method performs comparably or slightly better than the baseline in selected configurations.

Overall, our sparse strategies achieve higher ASR and sparsity while maintaining comparable $\ell_2$ perturbation magnitudes to the baseline, demonstrating clear advantages in resource-constrained attack scenarios.

### 4.4 Classifier Robustness under Sparse Attacks

To uncover structural factors behind model vulnerability, we further analyse the robustness of different architectures under varying sparsity levels.

As shown in Fig. 1c, ST-GCN Yan et al. (2018) consistently exhibits the highest susceptibility to sparse perturbations across all sparsity thresholds, followed by CTR-GCN Chen et al. (2021), while SkateFormer Do & Kim (2024) demonstrates the strongest resilience. Notably, even under extreme sparsity (e.g., $s = 0.95$), ST-GCN maintains a targeted attack success rate of approximately 48.5%, whereas the success rates of CTR-GCN and SkateFormer drop sharply to near zero. Although CTR-GCN and SkateFormer are based on fundamentally different architectures—GCN and Transformer, respectively—they exhibit similar resistance to highly sparse perturbations. Given that both incorporate cross-frame modelling and global information integration mechanisms, such structural characteristics may contribute to their improved tolerance against localised adversarial noise.

In contrast, under more relaxed sparsity constraints (*e.g.*, $s$ between 0.75 and 0.85), both ST-GCN and CTR-GCN achieve success rates above 80%, while SkateFormer maintains significantly lower vulnerability, with success rates remaining below 60%. This further highlights their architectural divergence: GCN-based models tend to focus on local neighbourhood information and are thus more susceptible to spatially localised perturbations, whereas the global attention mechanism in Transformer architectures leads to a more uniformly distributed representation across the sequence, improving robustness against such noise.

Moreover, Table 3 compares two spatial partition strategies, revealing model sensitivity to region selection. Under identical sparsity constraints, the global-saliency-based strategy consistently outperforms the local-structure partitioning across all models. This advantage is especially pronounced for SkateFormer, suggesting that Transformer-based models are particularly vulnerable to perturbations applied at globally salient joints.

### 4.5 Evaluation under Label-only Constraints

Under the strict label-only black-box setting, the attacker only has access to predicted labels without confidence scores, making it harder to identify effective perturbation regions. Without confidence-based saliency cues, the selection of critical frames and joints becomes less accurate, leading to suboptimal perturbation allocation.

This limitation affects architectures differently. ST-GCN Yan et al. (2018), with its localised receptive fields, can still propagate perturbations effectively even when the chosen directions are not optimal, thereby maintaining relatively high attack performance. In contrast, CTR-GCN Chen et al. (2021) and SkateFormer Do & Kim (2024) rely heavily on global aggregation mechanisms; under these conditions, inaccurately placed local perturbations exert limited influence on overall predictions, leading to more pronounced performance degradation.

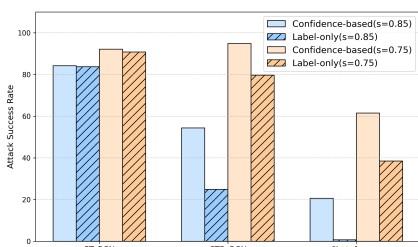

Figure 4: Targeted attack success rates across three model architectures under two access settings: confidence-based and label-only.

As illustrated in Fig. 4, these results suggest that models with stronger global modelling capabilities exhibit greater robustness under information-constrained attacks, whereas local models remain vulnerable in such scenarios.

## 5 Conclusion

We propose a saliency-guided spatio-temporal sparse black-box attack framework that achieves strong adversarial performance under minimal perturbation. Extensive experiments demonstrate its consistent superiority over baseline methods across various models and settings, including under high sparsity and label-only constraints. The results reveal distinct model vulnerabilities to sparse perturbations, offering new insights into the robustness of action recognition systems.

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

## A    APPENDIX

### A.1    FULL ALGORITHMIC WORKFLOW OF THE SPARSE ADVERSARIAL FRAMEWORK

---

**Algorithm 2** Saliency-Guided Spatiotemporal Sparse Attack (Targeted Setting)

---

1: **Input:** Targeted model $F$, original skeleton $x_0$, target class $y_{\text{adv}}$, frame segment length $l_f$, sequence length $T_f$, number of key regions $l_r$, total regions $T_r$, step size $u$, maximum steps $I_{\max}$

2: **Parameter:** Perturbation threshold $w$

3: **Output:** Adversarial example $x_{\text{adv}}$

4: $\hat{x} \leftarrow$ a sample from class $y_{\text{adv}}$                      // Sample from target class

5: $x_{\text{temp}} \leftarrow x_0,\ t \leftarrow 0$

6: $M_1 \leftarrow \text{FSIA}(F, x_{\text{temp}}, y_{\text{adv}}, l_f, T_f)$            // Frame-Segment Importance Assessment

7: $M_2 \leftarrow \text{SRIA}(F, x_{\text{temp}}, y_{\text{adv}}, l_r, T_r)$            // Spatial-Region Importance Assessment

8: $M \leftarrow M_1 \wedge M_2$                           // Element-wise AND of two masks

9: **while** $F.\text{predict\_label}(x_{\text{temp}}) \neq y_{\text{adv}}$ **do**

10:      $t \leftarrow t + 1$

11:      // Baseline Method: Periodic Frame Segment Masking

12:      // $M \leftarrow \text{zeros\_like}(x_0)$

13:      // $i \leftarrow t \bmod (T_f/l_f)$

14:      // $segment \leftarrow [i, i+1, \ldots, i+l_f - 1]$

15:      // $M[:, segment, :, :] \leftarrow 1$

16:      $\lambda \leftarrow \|\hat{x} - x_{\text{temp}}\|_2$

17:      $\delta \leftarrow (\hat{x} - x_{\text{temp}})/\lambda$

18:      $x_p \leftarrow x_{\text{temp}} + \lambda \cdot \delta \cdot M$

19:      $conf \leftarrow F.\text{predict\_confidence}(x_p, y_{\text{adv}})$

20:      **while** True **do**

21:          $\lambda \leftarrow u \cdot \lambda$                      //$u$ serves to increase the value of $\lambda$ur

22:          $x_p \leftarrow x_0 + \lambda \cdot \delta \cdot M$

23:          **if** $\text{SFP}(x_p) > w$ **then**

24:              $\lambda \leftarrow \lambda/u$

25:              **break**

26:          **end if**

27:          $new\_conf \leftarrow F.\text{predict\_confidence}(x_p, y_{\text{adv}})$

28:          **if** $new\_conf > conf$ **then**

29:              $conf \leftarrow new\_conf$

30:          **else**

31:              $\lambda \leftarrow \lambda/u$

32:              **break**

33:          **end if**

34:      **end while**

35:      $x_{\text{temp}} \leftarrow x_{\text{temp}} + \lambda \cdot \delta \cdot M$

36:      **if** $t > I_{\max}$ **then**

37:          **break**

38:      **end if**

39: **end while**

40: **if** $\text{SFP}(x_{\text{temp}}) > w$ **then**

41:      **return** Failure

42: **end if**

43: $x_{\text{adv}} \leftarrow \text{SIO}(x_{\text{temp}})$                  //SIO denotes Swarm Intelligence Optimization

44: **return** $x_{\text{adv}}$

---

**Perturbation Injection Procedure.**    In our sparse attack framework, adversarial perturbations are injected into critical regions of the input based on saliency-guided strategies. The detailed procedure is as follows:

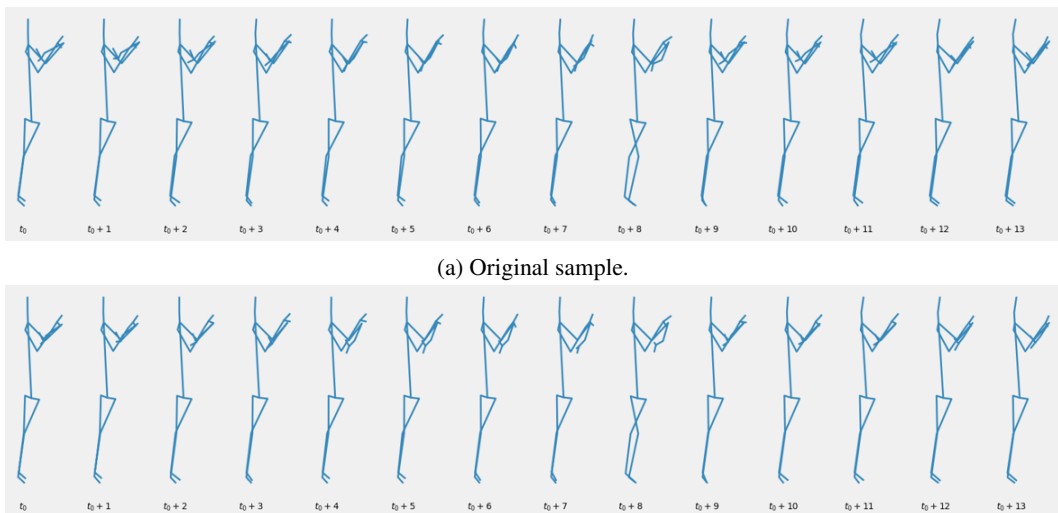

(a) Original sample.

(b) Adversarially perturbed sample.

Figure 5: Comparison between a clean input sample (a) and its adversarially perturbed counterpart (b). The perturbation leads to misclassification by the model, while preserving the natural appearance of the motion sequence.

In the **targeted attack** setting, we first sample a skeleton sequence $\hat{x}$ from the target class as the adversarial objective. In the **untargeted attack** setting, $\hat{x}$ can be sampled from any class different from the model's current prediction.

The attack direction is defined as $\hat{x} - x$, and its $\ell_2$ norm is computed as $\lambda$. The direction vector is then normalized to obtain the unit direction $\delta = \frac{\hat{x}-x}{\lambda}$.

Next, we apply our proposed sparsity selection strategy—composed of FSIA and SRIA—to obtain a spatio-temporal binary mask $M$, where each element is either $0$ or $1$, indicating whether perturbations will be applied to the corresponding region.

The perturbation is iteratively injected as follows: we initialize the step size $p = \lambda$, and at each iteration, we construct an adversarial example according to:

$$x_p = x_{\text{temp}} + p \cdot \delta \cdot M$$

where $x_{\text{temp}}$ denotes the current sample being optimized (initialized as $x_0$).

To control the perturbation strength, we iteratively scale $p$ by a factor $u > 1$ (i.e., $p \leftarrow u \cdot p$), and query the model for the confidence score of the target class. If the confidence increases, we continue enlarging the perturbation. If the confidence drops, we revert to the previous step size, update $x_{\text{temp}}$, and recompute the attack direction, repeating the process.

This procedure terminates when one of the following conditions is met:

- The model predicts the target class (attack success);
- The maximum number of iterations $I_{\max}$ is reached;
- The total perturbation exceeds the threshold $w$ (attack failure).

## A.2 VISUALIZATION OF PERTURBED SAMPLES

Figure 5 provides a visual comparison between a clean motion segment and its adversarially perturbed counterpart. Subfigure 5a depicts a temporal slice from the *clapping* action that is correctly classified by the ST-GCN model. Subfigure 5b shows the corresponding adversarial version generated by our sparse black-box attack framework.

The visualized segment was selected by our saliency-guided sparsity strategy as one of the key regions for perturbation. Consequently, every joint in each frame within this visualized segment

is perturbed. Despite this full perturbation within the selected region, the changes remain visually subtle and structurally coherent, preserving the natural appearance of the motion. Perturbations are primarily concentrated in the upper limbs—arms and hands—while maintaining overall body consistency.

The perturbed sequence leads the model to misclassify the action as *headache*, an unrelated class. This example demonstrates that our method can induce misclassification through sparse, localized, yet effective perturbations, highlighting both the stealth and spatiotemporal efficiency of the proposed attack.

### A.3 SWARM INTELLIGENCE OPTIMIZATION (SIO) MODULE.

To enhance the imperceptibility of adversarial examples, we introduce a **Swarm Intelligence Optimization (SIO)** module at the final stage of our attack pipeline. This module leverages the Particle Swarm Optimization (PSO) algorithm to refine the initial adversarial examples generated in previous steps.

The SIO module aims to produce adversarial examples that (1) maintain the attack success rate, (2) minimize the overall perturbation magnitude, and (3) preserve structural similarity to the original skeleton sequence. To this end, we design a fitness function that guides the optimization:

$$\mathcal{F}(x_{\text{adv}}) = -\alpha \cdot \text{ASR}(x_{\text{adv}}) + \beta \cdot \|\Delta x\|_2 + \gamma \cdot \text{JointDist}(x_{\text{adv}}, x_0)$$

where:

- $\text{ASR}(x_{\text{adv}})$ denotes the attack success indicator or a confidence-based surrogate;
- $\|\Delta x\|_2$ is the $\ell_2$ norm of the perturbation;
- $\text{JointDist}(x_{\text{adv}}, x_0)$ measures the average Euclidean distance between corresponding joints in the adversarial and original skeletons;
- $\alpha$, $\beta$, and $\gamma$ are hyperparameters that balance effectiveness, perturbation magnitude, and structural similarity.

Each particle in the swarm represents a candidate perturbation, and the fitness function guides the search towards perturbations that are both subtle and effective. Importantly, the SIO module plays an essential role in our framework: although the saliency-guided perturbation strategy can generate successful attacks, the resulting examples may still contain visually abrupt or spatially misaligned joints. SIO addresses this limitation by refining these initial adversarial samples into more natural and imperceptible forms.

### A.4 LIMITATIONS & BROADER IMPACTS

Despite the strong performance of our proposed sparse black-box attack framework across various models and settings, several limitations remain.

First, the initialization of the attack relies on reference samples from the target class to construct the perturbation direction. In some practical scenarios, such samples may not be accessible to the attacker, which limits the applicability of our method.

Second, under high sparsity constraints, the perturbation space is significantly reduced, making the optimization process more prone to local optima and increasing the difficulty of locating effective perturbation regions. This often results in substantially higher query costs.

Third, in the label-only setting, the absence of confidence feedback prevents adaptive adjustment of the perturbation direction, which reduces both the controllability and convergence efficiency of the attack.

Fourth, although our method achieves strong imperceptibility in most settings through sparse injection strategies, extremely high sparsity levels may require larger perturbation magnitudes to maintain attack effectiveness, which can partially compromise stealth.

Finally, our approach falls under the category of purely query-based black-box attacks, for which existing methods are extremely limited. The only known comparable baseline is BASAR. However,

due to a lack of clear implementation details in the original paper, we were unable to successfully reproduce its results on the NTU dataset, preventing a direct quantitative comparison. To provide a reasonable point of reference, we instead use a full attack version of our own method—without sparsity constraints—as the baseline. Although our sparse strategy achieves promising attack success rates, it appears to incur more queries than the full attack baseline. Nevertheless, we have run the publicly available BASAR demo (on the HDM05 dataset), which relies on geometric space transformation. From a qualitative standpoint, our method demonstrates comparable efficiency, though we regret that a direct comparison under a unified evaluation setting is currently unavailable.

Overall, our work aims to deepen the understanding of model vulnerabilities under sparse black-box attack settings. At the same time, we acknowledge that the proposed framework could potentially be misused to circumvent action recognition systems deployed in safety-critical applications such as security surveillance or healthcare.

We stress that our intention is to support the development of more robust models by identifying vulnerabilities, and to encourage the responsible use of adversarial techniques for safety evaluation. We hope that future research will integrate sparse attack strategies into standard robustness benchmarks and explore more effective defenses to strengthen existing systems.

## A.5  DISCUSSION AND FUTURE DIRECTIONS

While our proposed sparse black-box attack framework demonstrates strong performance across various action recognition models and settings, there remain several important aspects that warrant further exploration.

First, our strategy for identifying critical frames and joints relies on explicit saliency scoring, implemented through frame-wise masking and spatial region masking to evaluate their impact on model predictions. Although this approach is intuitive and broadly applicable, it incurs considerable computational overhead, especially when applied to long input sequences. Future work may explore more efficient region selection mechanisms, such as temporal neighborhood interpolation, frame difference aggregation, lightweight attention-based estimation, or reinforcement learning-based query strategies, to reduce redundant evaluations and improve key region extraction efficiency.

Second, the localized nature of sparse attacks, while enhancing imperceptibility, may render them more susceptible to defensive mechanisms. On one hand, robust recognition models with localized anomaly detection capabilities may effectively identify inputs with concentrated perturbations. On the other hand, certain defenses, such as temporal consistency checks or skeleton trajectory modeling, can exhibit stronger recovery performance under sparse manipulations. Therefore, future research should systematically evaluate the effectiveness of sparse attacks under diverse defense strategies, fostering the development of more resilient and generalizable attack techniques.

Third, in the label-only black-box setting, the absence of confidence feedback restricts the ability to iteratively refine the perturbation direction, thus limiting controllability and convergence. Although we construct the initial direction based on a reference sample, the inability to adjust direction based on model feedback remains a key limitation. One potential avenue is to leverage indirect signals, such as the temporal trend of predicted labels, success rates over fixed intervals, or custom-designed triggers based on reinforcement search to determine when to update the direction. These strategies could enable more adaptive perturbation refinement even in the absence of explicit confidence scores.

In summary, our work provides a novel perspective and implementation framework for sparse black-box attacks in skeleton-based action recognition. We hope the above discussions can inspire future research toward developing more efficient, robust, and practical adversarial strategies.

## A.6  LICENSES FOR USED ASSETS

- **NTU RGB+D 60/120 Datasets Shahroudy et al. (2016); Liu et al. (2019)** Freely available for academic use. License: Custom (research-only). Github: `https://github.com/shahroudy/NTURGB-D`

- **ST-GCN Yan et al. (2018)** Code licensed under MIT License. GitHub: `https://github.com/yysijie/st-gcn`

- **CTR-GCN Chen et al. (2021)** Code licensed under Apache 2.0. GitHub: `https://github.com/Uason-Chen/CTR-GCN`
- **SkateFormer Do & Kim (2024)** Code licensed under MIT License. GitHub: `https://github.com/KAIST-VICLab/SkateFormer`

## A.7  COMPUTATIONAL RESOURCES

All experiments were conducted on a local workstation equipped with four NVIDIA GeForce RTX 4090 GPUs (each with 24 GB of memory) and an Intel Xeon Gold 6430 CPU with 512 GB RAM. The software environment was configured using Ubuntu 22.04 LTS, PyTorch 2.1.1 with CUDA 12.1 support, Torchvision 0.16.1, and other dependencies as specified in the accompanying code repository.

The entire experimental process lasted approximately one month. We estimate the total compute usage to be around 2,500 GPU-hours (4 GPUs $\times$ 24 hours/day $\times$ 30 days), which fully covers all experiments reported in the paper. This includes training baseline models, performing adversarial attack evaluations, and running ablation studies.

To ensure reproducibility, we fixed random seeds across Python, NumPy, and PyTorch and provide the complete code and configuration files in the supplementary material and GitHub repository.

We acknowledge that the paper does not report variance measures (e.g., standard deviation, confidence intervals) or statistical significance tests. This decision was driven by the high computational cost of our sparse black-box attack framework: each attack involves thousands of model queries under strict sparsity constraints, making repeated trials impractical within our resource budget. Furthermore, the attack procedure is deterministic with fixed hyperparameters, and the results are highly consistent across large datasets (NTU RGB+D 60 and NTU RGB+D 120) and multiple models. Therefore, while statistical tests were omitted, the observed performance trends remain stable and reliable.

