# OpenReview forum: "Sparse but Sharp: Saliency-guided Black-box Attacks  Reveal Vulnerabilities in Skeleton-based Human  Action Recognition"
_ICLR.cc/2026/Conference — Submitted to ICLR 2026_

### Official Review · Reviewer_g7yA · 2025-10-29

**Soundness:** 2
**Presentation:** 2
**Contribution:** 2
**Rating:** 4
**Confidence:** 3

**Summary:**

This paper proposes a saliency-guided spatio-temporal sparse black-box attack framework targeting skeleton-based human action recognition models. By selectively perturbing critical frames and joints identified through temporal and spatial saliency analysis, the method achieves high attack success rates with minimal perturbations, enhancing stealth and efficiency. Extensive experiments on NTU datasets demonstrate the framework's effectiveness.

**Strengths:**

(1) This paper proposes a saliency-guided joint selection method that accurately detects the most impactful spatio-temporal regions without relying on gradient-based information.
(2) This paper provides a comprehensive analysis of the adversarial vulnerabilities of GCN-based and Transformer-based models, highlighting their distinct sensitivities to spatio-temporal perturbations.

**Weaknesses:**

(1) While the proposed spatio-temporal sparse attack framework (Temporal Sparsity and Spatial Sparsity) incorporates saliency analysis, the concept of sparsity itself is not new. Sparse attacks have been widely studied, especially in the image domain. The paper does not clearly articulate how its method is technically superior or innovative compared to existing approaches.
(2) Although the paper mentions prior methods like BASAR and TASAR, it does not thoroughly discuss the differences between these methods and the proposed approach. For instance, BASAR is referenced, but its implementation could not be reproduced, leading to a lack of direct comparison, which weakens the credibility of the experimental evaluation.
(3) The paper lacks diversity in both datasets and model selection, as the experiments are conducted only on NTU 60 and NTU 120, which, while widely used, offer limited scene and action diversity. Additionally, the experiments focus on three models (ST-GCN, CTR-GCN, and SkateFormer), with ST-GCN receiving the most attention, while newer skeleton-based action recognition models, such as those based on hybrid architectures or self-supervised learning, are not included, restricting the generalizability and applicability of the proposed framework to more complex datasets and recent advancements in the field.

**Questions:**

(1) The saliency analysis relies on masking frames and joints to evaluate their importance. However, in the confidence-based black-box setting, model outputs may not be sensitive to minor perturbations, which could lead to inaccurate saliency assessments. How does the framework handle cases where the saliency analysis is unreliable?
(2) The paper mentions that increasing sparsity reduces the attack success rate, but it does not quantify the relationship between sparsity (e.g., the proportion of selected frames and joints) and performance. How does the framework determine the optimal balance between sparsity and attack success? Additionally, does the sparsity strategy need to be adjusted for different models?
(3) Are the adversarial examples generated by the sparse attack transferable across models? For instance, can an example crafted for ST-GCN successfully attack CTR-GCN or SkateFormer? The paper does not evaluate the cross-model transferability of its adversarial examples.

---

> ### Author Response · Authors · 2025-11-17
>
> **We sincerely thank the reviewer for the insightful and constructive comments. Below we respond to each point in order.**
>
> ------
>
> ## **Q1: Question on the reliability of saliency analysis.**
>
> **A1:**
>  In the confidence-available black-box setting, we adopt the mask-based **Frame Segment Importance Assessment (FSIA)** and **Spatial Region Importance Assessment (SRIA)** mechanisms. Specifically, we mask different temporal segments and spatial regions, then measure the corresponding decrease in the target-class confidence score to rank their importance (see Sections 3.1 and 3.2 of the paper, as well as the accompanying algorithm pseudocode). In practice, to mitigate potential unreliability in saliency estimation, we **select several of the most critical spatiotemporal segments from the ranking results—not just a single one—and apply perturbation injection to each of these segments individually**, thereby achieving more effective attack performance.
>
> Furthermore, to address the possibility that the saliency assessment may still exhibit instability, our algorithm incorporates a **re-estimation / re-selection** mechanism (Algorithm 2), which re-evaluates perturbation directions and key regions when necessary. This process helps correct deviations that may arise from inaccurate saliency estimation. In addition, to enhance perturbation effectiveness and attack stability, we apply a **Swarm Intelligence Optimization (SIO)** module after perturbation injection to perform fine-grained adjustment of the perturbations.
>
> ------
>
> ## **Q2: Question on the relationship between sparsity strategy and performance.**
>
>
> **A2:**
>  The paper already includes several quantitative results demonstrating a clear relationship between sparsity, attack success rate (ASR), and query cost. In the temporal sparsity ablation study (Table 1), targeted ASR decreases substantially as sparsity increases from $s=0.0$ to $s=0.997$: specifically, targeted ASR drops from **95.32%** at $s = 0.0$ to **5.60%** at $s = 0.997$, while the query cost increases correspondingly. This provides direct evidence of the trade-off between sparsity and attack success. Spatial sparsity (Table 2) and joint spatio-temporal sparsity (Table 3) further reinforce this observation: in the **medium-sparsity range** (e.g., $ \approx 0.75 \text{–} 0.85$), the attack achieves a favorable balance between ASR and imperceptibility, whereas extremely high sparsity significantly reduces targeted ASR but has a smaller impact on untargeted attacks.
>
> Our experiments also reveal a relatively stable sparsity interval—represented in the paper through settings such as **T1/T2/T3** and **S1/S2/S3**—within which different models (ST-GCN and CTR-GCN) generally maintain a reasonable balance between ASR and stealthiness. In contrast, **SkateFormer**, being more robust under the same sparsity levels, requires **lower local sparsity** to achieve comparable attack performance. These findings indicate that certain models are indeed more sensitive to sparsity, and slight adjustments to the sparsity strategy are needed to obtain optimal results across architectures.
>
> ------
>
> ## **Q3: Question on cross-model transferability.**
>
>
> **A3:**
>  In Sections 4.3–4.4 of our paper, we have already conducted a comparative analysis of the vulnerability of three models (ST-GCN, CTR-GCN, and SkateFormer) under the same attack settings, and discussed the robustness differences exhibited by different architectures under sparse attacks. However, we sincerely acknowledge that the current version of the paper does not include the typical cross-model transferability experiments, i.e., generating adversarial examples on model A and transferring them to model B.
>
> Conducting transfer attacks generally requires generating a large number of adversarial samples for each model pair (A→B) and performing cross-model evaluation. Under high sparsity attack settings, each attack entails a high number of queries (QC) and substantial computational cost. We have also explicitly stated in the appendix the limitations of our current study in terms of computational resources. Furthermore, different models have significant variations in input frame numbers—for instance, ST-GCN takes 300 frames as input, while CTR-GCN only takes 64 frames. Since our method relies on saliency guidance to apply sparse perturbations at critical time steps, the perturbation positions have clear temporal specificity. As a result, under inconsistent input frame numbers, adversarial perturbations cannot be directly aligned or mapped to another model’s input structure, which poses additional challenges for transferability experiments.
>
> Nevertheless, we greatly appreciate the reviewer’s suggestion. We agree that cross-model transferability analysis is indeed of significant importance, and we will consider this direction as a priority in future work to further deepen our understanding of the shared vulnerabilities across different architectures.

---

### Official Review · Reviewer_cAET · 2025-10-31

**Soundness:** 3
**Presentation:** 3
**Contribution:** 3
**Rating:** 4
**Confidence:** 4

**Summary:**

This paper proposes a spatiotemporally sparse black-box attack method for human action recognition. By selecting the most salient temporal segments and joint positions in both time and space, the method perturbs the input sequence minimally while still achieving effective attacks. Experimental results show that under a limited perturbation budget, the proposed approach achieves a high attack success rate.

**Strengths:**

1. The spatiotemporal sparsity focuses effectively on key temporal frames and joint positions.
2. The attack is validated on both GCN-based and Transformer-based recognition models.
3. The experiments take realistic perturbation budgets into account, and the results are quite promising.

**Weaknesses:**

1. The construction of spatiotemporal sparsity relies on the confidence scores provided by the target model.
2. In settings where only labels are available, the method depends on predefined masks and attenuation factors, which may limit the diversity of generated adversarial samples.
3. For models that rely on global information, such as SkateFormer, the attack performance drops significantly.

**Questions:**

1. Perturbation budget: What do the budget values (3000 and 15000 for NTU60 and NTU120) represent, and how are they determined?
2. Effect of QC: Under the same budget, is the higher performance compared with the baseline partly due to the sparsity in frames and joints, which may save budget and allow for more query counts (QC)?
3. Comparison with other methods: The authors mention that their baseline represents only a lower bound. Are there comparisons with other SOTA methods in terms of performance and budget?

---

> ### Author Response · Authors · 2025-11-17
>
> **We sincerely thank the reviewer for the insightful and constructive comments. Below we respond to each point in order.**
>
> ------
>
> ## **Q1: Perturbation budget**
>
> **What do the budget values (3000 and 15000 for NTU60 and NTU120) represent, and how are they determined?**
>
> **A1:**
>  Regarding the question about the perturbation budget (3000 and 15000 for NTU60 and NTU120), these values are described in Section 4.3 (line 413) of the paper as: "For ST-GCN, which processes the full 300-frame sequences, we set the total perturbation budgets to 3000 and 15000 for NTU60 and NTU120, respectively." These values (3000 and 15000) represent the total perturbation budget.
>
> The total perturbation budget refers to the maximum total perturbation energy allowed during the adversarial attack. When perturbations are added to a skeleton sequence, this total perturbation budget defines the upper limit of the perturbation energy we can apply. If the attack is not successful within this budget, the attack is considered to have failed. This total perturbation energy is related to the Single-Frame Perturbation (SFP) defined in Section 3.3. To improve the stealthiness of the attack, we impose a constraint that the perturbation per frame should not exceed the SFP. Thus, the total perturbation budget is calculated as SFP × number of frames. For instance, for ST-GCN with 300 frames, if we set SFP to 10, the total perturbation budget would be $10 \times 300 = 3000.$
>
> As for the SFP, it is defined as the maximum allowable perturbation per frame and is a constant value based on our perturbation injection method. We first compute the perturbation guidance signal $\delta = \hat{x} - x_0$, then normalize it to obtain the unit tensor of the perturbation direction. We multiply this unit tensor by a constant and add it to $x_0$, gradually increasing the constant until the attack succeeds or the SFP is reached.
>
> ------
>
> ## **Q2: Effect of QC**
>
> **Under the same budget, is the higher performance compared with the baseline partly due to the sparsity in frames and joints, which may save budget and allow for more query counts (QC)?**
>
> **A2:**
>  Your understanding is absolutely correct. Specifically, our proposed sparse attack method first locates key regions based on confidence scores and then injects perturbations into these critical regions to perform the attack. Under the same perturbation budget, our method achieves a higher attack success rate compared to the baseline. We attribute this improvement to the algorithm’s ability to successfully identify and target these key regions, while the baseline method performs a full-scale attack without focusing on specific key regions. The increase in query counts (QC) is primarily due to the process of identifying and targeting these critical regions.
>
> ------
>
> ## **Q3: Comparison with other methods**
>
> **The authors mention that their baseline represents only a lower bound. Are there comparisons with other SOTA methods in terms of performance and budget?**
>
> **A3:**
>  The only known comparable baseline is BASAR (Diao et al., CVPR 2021). However, due to the insufficient implementation details provided in the original paper, we were unable to successfully reproduce its results on the NTU datasets, which prevents a direct quantitative comparison. To offer a reasonable and fair reference point, we therefore adopt the full-attack version of our own method (without sparsity constraints) as the baseline.
>
> In addition, we have also tested the publicly available BASAR demo (implemented on the HDM05 dataset), which is based on geometric space transformation. From a qualitative standpoint, our method exhibits comparable efficiency. Nevertheless, we regret that a direct and comprehensive comparison under a unified evaluation protocol is currently not feasible.

---

### Official Review · Reviewer_c3Vh · 2025-11-02

**Soundness:** 2
**Presentation:** 2
**Contribution:** 2
**Rating:** 4
**Confidence:** 4

**Summary:**

This paper proposes a saliency-guided spatio-temporal sparse black-box adversarial attack framework tailored for skeleton-based human action recognition systems. The method identifies critical joints and temporal segments through heuristic saliency estimation—without access to model gradients—and injects perturbations only into these localized regions. The approach is evaluated under both confidence-based and label-only black-box settings across multiple state-of-the-art models (e.g., ST-GCN, CTR-GCN, SkateFormer) on the NTU60 and NTU120 datasets. Results show that the proposed attack achieves high attack success rates with significantly reduced perturbation budgets, demonstrating superior stealth and query efficiency compared to dense baselines. The work also provides insights into the differing robustness of GCN- versus Transformer-based architectures under sparse perturbations.

**Strengths:**

1. Comprehensive Evaluation: The paper includes thorough experiments across multiple models, datasets, and attack settings (untargeted/targeted, confidence-based/label-only), offering valuable empirical insights.

2. Imperceptibility Emphasis: The use of spatial-temporal sparsity and the SIO (Swarm Intelligence Optimization) module enhances the naturalness and stealth of adversarial examples.

**Weaknesses:**

1. Limited Practicality of Perturbation Model:
The paper assumes that an attacker can directly add noise to 3D joint coordinates, which may not reflect real-world deployment. In practice, skeleton sequences are typically estimated from RGB or depth sensors (e.g., via pose estimators like OpenPose or MediaPipe). A more realistic threat model would involve physical perturbations—such as wearable patches or clothing—that indirectly alter joint positions. Without addressing how such perturbations translate into skeletal noise (e.g., through a pose estimation pipeline), the attack’s applicability remains largely theoretical.
2. Lack of Technical Novelty:
While the application to skeleton-based action recognition is timely, the core idea—sparse, saliency-guided adversarial perturbations in spatio-temporal domains—has been explored in prior video attack literature. For instance, Wei et al. (AAAI 2019), “Sparse Adversarial Perturbations for Videos” already introduced content-aware sparse attacks on video classifiers. The current work adapts similar principles (temporal segmentation, spatial region selection, mask-based importance scoring) to skeletons but does not sufficiently differentiate its methodological contribution or analyze why skeletons demand a fundamentally new approach.
3. Diminishing Returns of Pure Attack Research:
Adversarial vulnerability in deep models is now a well-established phenomenon. Given the maturity of the field, the community increasingly prioritizes robustness-enhancing defenses, certified robustness, or attack-aware training. This paper, while technically sound, contributes another attack without proposing mitigation strategies or actionable guidance for defenders. A more impactful direction would be to leverage the identified vulnerabilities to design robust architectures (e.g., saliency-regularized training or sparsity-aware data augmentation).

**Questions:**

see the weakness

---

> ### Author Response · Authors · 2025-11-17
>
> **We sincerely thank the reviewer for the insightful and constructive comments. Below we respond to each point in order.**
>
> **Reviewer Concern 1:**
>  The paper assumes direct perturbation on 3D joint coordinates, which may not reflect real-world deployment. In practice, skeletons are estimated from RGB/depth sensors, and realistic attacks may involve physical perturbations. Without modeling how such physical perturbations propagate through pose estimation, the attack’s applicability remains theoretical.
>
> **A1**: Your point is completely valid, and we sincerely appreciate this important observation. However, we would like to clarify that adversarial attack research in skeleton-based action recognition traditionally focuses on perturbations applied **directly to skeleton sequences**, rather than on the accuracy or robustness of the pose estimation process. Most existing skeleton-based action recognition models are trained using clean and known 3D skeleton data, typically under the assumption that such data are accurate. Therefore, our experimental setting follows this widely adopted convention and assumes that the skeleton data produced by pose estimation is correct.
>
> Regarding the physical perturbations you mentioned (e.g., wearable patches or clothing), we fully agree that they represent realistic and meaningful research directions. However, such perturbations belong primarily to the domain of **pose estimation attacks**, which is beyond the scope of skeleton-based action recognition. Our focus is on analyzing the vulnerability of recognition models given skeleton inputs. Under this setting, we propose a black-box attack and further extend it to a **label-only** setting, avoiding reliance on confidence scores. This extension is particularly valuable in practical scenarios where detailed model outputs are unavailable.
>
> ------
>
> **Reviewer Concern 2:**
>  The core idea—sparse, saliency-guided spatiotemporal perturbations—has been studied in prior video attack work (e.g., Wei et al., AAAI 2019). The current work applies similar ideas to skeletons without sufficiently differentiating its contributions or explaining why skeletons require a new approach.
>
> **A2:** Thank you for raising this thoughtful comment. Our sparse attack method is not proposed merely as a new attack technique, but more importantly as a tool for **analyzing robustness differences among different skeleton-based action recognition architectures**, particularly in terms of their attention to different spatiotemporal regions. This analytical perspective—extensively discussed in Section 4—constitutes a key contribution of our work.
>
> While we acknowledge that there are conceptual similarities to Wei et al. (AAAI 2019), it is important to emphasize that skeleton data and video data differ fundamentally. Wei et al.’s method operates on short video clips with only 16 frames, whereas skeleton sequences typically contain around **300 frames**, leading to far greater spatiotemporal complexity. As a result, directly applying their method to skeleton data would be inappropriate. Our work is the **first** to study sparse black-box adversarial attacks specifically in the skeleton-based action recognition domain, and we further incorporate **Swarm Intelligence Optimization (SIO)** to refine the search for optimal adversarial samples—an innovation unique to our approach.
>
> Additionally, our extension to a **label-only** setting significantly enhances the real-world applicability of the method, as it does not rely on model confidence scores. Thus, despite conceptual connections to Wei et al., our work differs substantially in application context, data characteristics, and algorithmic design, demonstrating its distinct contributions.
>
> ------
>
> **Reviewer Concern 3:**
>  Adversarial vulnerability is well-known, and the community is increasingly interested in robustness and defenses. The paper proposes another attack without offering mitigation strategies or advice.
>
> **A3:** Thank you very much for this valuable perspective. We fully agree that adversarial vulnerability is a well-established phenomenon and that current research trends emphasize robustness enhancement and defense strategies. However, the motivation behind our work is to investigate **how different model architectures vary in their robustness**, and our sparse attack serves as an analytical tool to reveal the specific spatiotemporal weaknesses of each architecture. This provides meaningful insights—beyond merely achieving an attack—that may help guide the design of more robust models. We hope this analytical contribution can inspire subsequent work on robustness-aware architectures.
>
> We acknowledge that our current work does not propose explicit defense mechanisms. This is indeed an important research direction, and we plan to explore it as part of our future work, leveraging the vulnerabilities identified in this study to develop corresponding defense strategies.

---

### Meta-Review · Area_Chair_CzPJ · 2026-01-06

**Summary:**

- The attack assumes direct access to and manipulation of 3D joint coordinates, which is unlikely in real-world deployments where skeletons are estimated indirectly from sensors via pose estimation pipelines.

- Sparse, saliency-guided adversarial attacks are well studied, particularly in video and image domains. The paper does not clearly articulate what is fundamentally new or skeleton-specific in its methodology.

- The experimental setup is considered incomplete: unclear perturbation budgets, lack of strong baselines or SOTA comparisons, and limited dataset/model diversity reduce confidence in the conclusions.

**Reviewer Concerns:**

- The authors justify using direct 3D-joint perturbations by citing convention in skeleton-based action recognition research, but this does not address the reviewer’s concern that real-world attacks operate on skeletons estimated from sensors. The applicability to realistic scenarios remains untested.

- The rebuttal claims the contribution is studying robustness differences across architectures, but this is an analytical use of the attack, not a new attack technique. The reviewer’s concern was about methodological originality, which remains unconvincing.

- The rebuttal partially justifies the chosen baseline, but it fails to provide the quantitative SOTA comparisons and cross-model evaluation, which the reviewers find important for comprehensive evaluation.

**Reviewer Scores:**

All reviewers are likely to maintain their current ratings.

---

### Decision · Program_Chairs · 2026-01-26

Reject